# Telemedical Consultations in Palliative Care: Benefits through Knowledge Exchange and Intercollegiate Collaboration—Findings from the German oVID Project

**DOI:** 10.3390/cancers15092512

**Published:** 2023-04-27

**Authors:** Andreas Bückmann, Florian Bernhardt, Maria Eveslage, Michael Storck, Gerold Thölking, Helga Buss, Dirk Domagk, Christian Juhra, Philipp Lenz

**Affiliations:** 1Department of Palliative Care, University Hospital Muenster, 48149 Muenster, Germany; andreas.bueckmann@ukmuenster.de (A.B.);; 2West German Cancer Center Consortium (WTZ), Network Partner Muenster, University Hospital Muenster, 48149 Muenster, Germany; 3Institute of Biostatistics and Clinical Research, University of Muenster, 48149 Muenster, Germany; 4Institute of Medical Informatics, University of Muenster, 48149 Muenster, Germany; 5Department of Internal Medicine and Nephrology, University Hospital Muenster Marienhospital Steinfurt, 48565 Steinfurt, Germany; 6Department of Geriatrics, EVK Muenster Alexianer Johannisstift GmbH, 48147 Muenster, Germany; 7Department of Medicine I: Internal Medicine and Gastroenterology, Josephs-Hospital Warendorf, Academic Teaching Hospital of the University of Muenster, 48231 Warendorf, Germany; domagkd@uni-muenster.de; 8Office for eHealth, University Hospital Muenster, 48149 Muenster, Germany

**Keywords:** palliative care, palliative medicine, telemedicine, telehealth, telemedical consultations, medical collaborations, interprofessionality, knowledge exchange

## Abstract

**Simple Summary:**

Both palliative care and telemedicine are emerging disciplines that were combined in the present study to transfer expertise between inpatient tertiary and primary care hospitals. In this context, our study focuses solely on the effects of telemedicine from the physicians’ perspective, so consultations were also permitted without direct patient involvement or reference. We were able to demonstrate that due to this lower threshold linkage, unnecessary referrals and transfers can be avoided while maintaining high satisfaction among the conducting physicians. Moreover, it facilitates the short-term incorporation of further specialized medical disciplines and intercollegiate collaboration. However, there are still numerous technical issues at present. But overall, telemedicine offers great advantages in palliative care, which ought to be exploited in the future.

**Abstract:**

(1) Background: As the number of people receiving specialized palliative care (PC) continues to rise, there is a need to ensure the transfer of this expertise from university-based PC departments to primary care hospitals without such in-house access. The present study examines the potential of telemedicine to bridge these gaps. (2) Methods: This is a prospective multi-center feasibility trial. All physicians were appropriately pre-equipped and instructed to conduct telemedical consultations (TCs), which took place within fixed meetings or on-call appointments either related or unrelated to individual patients (allowing TCs also for educational and knowledge exchange purposes). (3) Results: An inquiry for participation was submitted to 11 hospitals, with 5 external hospitals actively cooperating. In the first study section, a total of 57 patient cases were included within 95 patient-related TCs during 80 meetings. Other university disciplines were involved in 21 meetings (26.2%). Therapy adjustments resulted following 25 of 71 affected TCs (35.2%). In 20 cases (21.1%), an on-site consultation at the university hospital was avoided, and in 12 cases (12.6%), a transfer was avoided. Overall, TCs were considered helpful in resolving issues for 97.9% of the cases (*n* = 93). Yet, technical problems arose in about one-third of all meetings for at least one physician (36.2%; *n* = 29). Besides, in the second study section, we also conducted 43 meetings between physicians for education and knowledge exchange only. (4) Conclusions: Telemedicine has the potential to transfer university expertise to external hospitals through simple means. It improves collaboration among physicians, may prevent unnecessary transfers or outpatient presentations, and is thus likely to lower costs.

## 1. Introduction

Even before COVID-19, the potential of telemedicine in palliative care (PC) was recognized, although the pandemic was a catalyst for forced trial-and-error approaches and learning quickly about the applicability of technology in healthcare [1]. There is already a wide variety of telemedical concepts, such as bundling of services, telegenetics, or telepathology, that are predominantly equivalent to face-to-face care. These are cost-effective and demonstrate high patient satisfaction [2,3]. However, telemedicine has been a fuzzy term till now, encompassing both synchronous communication tools such as telemedical consultations (TCs) as used in the present study, and also asynchronous tools such as digital messages or websites [4]. In particular, patients’ previous experiences with technology can contribute to a positive attitude towards it, which is why initial contact should be sought early in the illness trajectory and can improve familiarity. Yet, it remains important that there is an opportunity to choose the preferred type of care [5].

The idea of digital-assisted integration of PC seems ideally applicable for a broad range of novel telehealth approaches, particularly since PC is increasingly based on the complexity of patients’ care and needs, irrespective of their prognosis [6]. PC, as defined by the World Health Organization, is an interdisciplinary field that aims to improve the lives of patients facing life-threatening diseases, and it is usually provided by specialized consultation teams or units in inpatient healthcare settings. Evidence points to PC as being associated with lower symptom burden, better quality of life, and no decreased survival [7,8,9,10]. Advanced cancer diagnoses account for the largest group of people receiving hospice care or PC. In fact, Hess et al. found that only 8.1% of their sample was diagnosed with non-cancer [11]. Common symptoms among PC patients diagnosed with advanced cancer involve fatigue, pain, dry mouth, loss of appetite, weight loss, and sleep problems [12]. Likewise, psychological symptoms are frequent among these patients [13]. Yet, the symptom burden of patients with both malignant and nonmalignant diseases is quite similar once adjusted for confounders [14]. Although PC is a well-established type of care in the contemporary German healthcare system, adequate coverage of patients with palliative needs is still not assumed. Studies describe potential opportunities for improvement through educational strategies, process mapping, feedback, multidisciplinary meetings, and multiple implementation strategies [15]. It has already been shown in several studies, recently also by our own working group, that collaboration between different departments with the PC unit is feasible and beneficial both for patient outcomes and the attending physicians within a single hospital [16]. Similarly, in the context of home-based PC patients, van Gurp et al. demonstrated benefits through collaboration between primary care and specialized PC physicians who used telemedicine for linkage [17]. However, there are also studies in outpatient settings that cast doubt on the benefits of TC when these are conducted directly between members of the PC team and their patients [18].

For patients, the feeling of safety is of high relevance when receiving a PC. According to Dillen et al., this includes the following topics: (i) patient-centeredness: availability, provision of information/education, professional competence, patient empowerment, and trust; (ii) organizational work: comprehensive responsibility, external collaboration, and internal cooperation; and (iii) direct communication [19]. Thus, both from providers’ and patients’ perspectives, increased collaboration between departments and healthcare providers is structurally advantageous and offers additional benefits within the care setting. In 2015, the “Hospice and Palliative Care Law” emphasized empowering and extending general PC in Germany [20]. In some areas, this has been followed by the expansion of inpatient departments and outpatient service providers, necessitating appropriate linkages and interactions between them. Recommendations for local collaboration thus address missions and aims, roles and responsibilities, coordination, communication and information channels, public visibility, and funding [21]. As can be seen from the current scientific status, besides immediate therapeutic and care benefits for individual patients, regular exchanges between PC providers are also likely to arise with advantages beyond that. Hence, both the potential impact of telemedicine on physicians in terms of improved treatment for patients in primary care hospitals, as well as on an intercollegiate “general” level and on knowledge exchange, were investigated. By recording avoided outpatient presentations and transfers to the tertiary care hospital, this study also sought to indirectly measure its potential impact on patients.

## 2. Materials and Methods

We performed a prospective multi-center feasibility trial (ethical approval: 2019-683-f-S “oVID—open video system in medicine”). Informed consent was obtained from all patients, and collaborations were established with each participating hospital and healthcare provider. The study protocol conformed to the ethical guidelines of the 1975 Declaration of Helsinki. This trial was conducted from February 2019 to February 2022, with recruitment lasting between January 2020 and October 2021.

The study was primarily supervised by the University Hospital of Muenster (a tertiary care hospital) and conducted in collaboration with a group of clinics and palliative health care providers within the same federal state. It was embedded in a major telemedicine project involving multiple departments. Oelmeier et al. from the Department of Gynecology and Obstetrics previously published a paper concerning a different arm of this project [22].

All TCs were held between senior physicians from each provider’s team—with the involvement of other university hospital departments as necessary—with or without the presence of the patients, and they were conducted online either upon request by one of the two healthcare providers with reference to a specified patient or for educational or information-sharing purposes. These two parts are presented separately in the results section. Throughout, we refer to the term “consultation” whenever we discuss a particular conversation about a patient or issue, and the term “meeting” whenever we report a continuous videoconference session. Accordingly, more than one consultation could be held within a single meeting, but not vice versa. Although PC always ought to be performed by a multi-professional team, in this study, we focused exclusively on communication between physicians. Besides dedicated third-party funding, this was also motivated by the fact that, particularly in smaller primary care hospitals, physicians often serve as the principal “managers” of patient cases and as multipliers of new scientific knowledge. However, additional disciplines and professions could be incorporated into the meetings at any time. Stationary computers or portable notebooks operating under Microsoft Windows 10 were used via a wired or wireless internet connection. Technically, the project is based on Web Real-Time Communication (WebRTC), which enables computer-to-computer connectivity. This is implemented using CGM ELVI, an end-to-end encrypted electronic video consultation system that supports audio, video, and text conversations, along with desktop and file sharing. Participating physicians were properly equipped and instructed in advance, enabling the independent performance of TCs. These were either conducted as part of fixed sessions or on a short-term, on-demand basis. More than one consultation during a meeting were allowed, depending on the simultaneous number of patients of the respective external hospital and their individual need for counseling.

For the evaluation of these sessions, participants had access to integrated questionnaires with predetermined response options or open-text answers embedded directly in ELVI. In addition, certain details of the consultations were documented by the physician based in the study center using the in-house electronic hospital information system (ORBIS by Dedalus). Descriptive data on the included patients were therefore collected via this system. In only a few individual cases, there were minor problems with the electronic documentation process through these programs, such that complete datasets were not available for all patients and sessions. This will be specified in greater detail within the limitations section; the corresponding data have been redacted. Similarly, some items were only displayed after conditional questions, resulting in differing reference sizes given in each case.

Inclusion criteria were a minimum age of 18 years, a progressive life-limiting disease, the capacity to adequately understand the study information, and being able to provide written informed consent. Any individuals unable to sufficiently understand the study information or provide informed consent were excluded. Alternatively, if a legal representative was appointed, that person was also allowed to provide informed consent for participation by proxy. All patients were selectively asked to participate within routine clinical practice if tertiary care consultations were required for them or if the attending physicians requested third-party assistance for their treatment.

The primary outcome of the first study section was the avoidance of face-to-face consultations at the university hospital by patients or the transfer of patients due to the complexity of their condition out of the cooperating healthcare institutions. These assessments were made jointly by the primary care and tertiary care physicians involved after each meeting. Moreover, the physicians’ level of satisfaction (on an ordinal scale ranging from 1 to 6) regarding the consultations was assessed, followed by an evaluation of the usefulness and the gain of knowledge. In addition, the occurrence and type of any technical problems during the meetings were recorded. Furthermore, the rationales for initial consultation requests and whether there had been resulting adjustments in therapy post consultations were documented. For the second study section, no primary or secondary outcomes were defined in advance, but physicians’ level of satisfaction and descriptive data were also recorded throughout.

Patients’ demographic data were analyzed using descriptive statistics. Continuous data are presented as the mean and their standard deviation (SD). Categorical data are presented as absolute and relative frequencies. Multiple answers were possible to the question about the reasons for a request. In order to quantify the degree to which reasons are associated with each other, the coincidence index according to Dice was used [23]. All statistics were performed using the program R version 4.1.0 [24].

## 3. Results

### 3.1. First Study Section: TCs with Direct Patient Reference

A request to participate in this study was sent to 11 hospitals, of which 5 regional hospitals (45.5%) were actively involved in its realization. According to our digital record, we enrolled 59 patients in this study. Yet, prior to analysis, 2 out of the 59 patients (3.4%) were excluded from analysis due to technical problems or incorrect assignment of their study section. Thus, a total of 57 persons with an average age of 69.7 years (SD: 13.6) were included in the TCs with direct patient reference. Of these, there were 34 (59.6%) females and 23 (40.4%) males. On average, the patients were located 20.6 km (SD: 21.6) from the university hospital, with the farthest distance from home to the hospital being 102 km. The majority of our cohort suffered from cancer (*n* = 46; 80.7%), which was predominantly metastatic (*n* = 39; 68.4%). Most of our patients (*n* = 24; 42.1%) experienced an unstable phase of their disease, while 16 patients (28.1%) showed a current worsening of their condition. An average of 12.5 treatment days (SD: 13.7) had passed by the time of inclusion during the current hospitalization. At this time, active treatment was still ongoing for 21 patients (36.8%; additionally, no statement was provided regarding this information for *n* = 9 patients (15.8%)), including radiation in 13 patients (22.8%), chemotherapy or immunotherapy in 25 patients (43.9%), and surgery in 21 patients (36.8%). Treatment limits had already been discussed and established among 27 patients (47.4%). According to the five-level German classification of the need for care, most of the patients belonged to level 3 (*n* = 13; 22.8%), although this information was not known to the physicians or apparent in the medical records for 24 patients (42.2%). More detailed information on the recruitment and study process can be found in Figure 1. Further details on patient characteristics are listed in Table 1.

A total of 95 TCs were carried out relating to the 57 enrolled patients, which were distributed over a total of 80 meetings. Of these patients, depending on the recording by the university hospital or collaborating parties, data on performed TCs were available in 54 and 53 cases, respectively. However, as the missing data were not related to identical patients, any lacking details were entirely supplemented by using each other’s clinic records subsequently. Patients were present during 34 visits (35.8%). These meetings lasted on average 23.1 min (SD: 14.7), with the longest meeting taking 73 min. In 21 meetings (26.2%), physicians from other departments also participated. The majority of requests for consultations were based on the need for general PC consultations (n = 22; 38.6%), but in 17 consultations, patients (29.8%) were specifically requested for transfer to or out of the university hospital. Since more than one reason for intercollegiate consultation was frequently mentioned for a single patient, we depicted the frequencies and the reasons given jointly in Figure 2 and Figure 3 (only 71 answers are considered here, as this answer was preceded by a conditional question). Overall, technical problems arose in 20.0% (*n* = 16) of the meetings for both physicians, and only one physician indicated such problems in 16.2% (*n* = 13). Following the consultations, adjustments in therapy or medication were made in 25 cases (35.2%), and in 33 cases (46.5%) other beneficial outcomes resulted following collaborative peer exchange regarding patient care. Further information concerning the meetings and consultations can be found in Table 2.

As a result of the TCs, a face-to-face presentation by the patient at the university hospital was prevented in 20 cases (21.1%). In addition, in 12 cases (12.6%), an otherwise intended transfer of patients from another hospital to the university hospital was avoided due to the online consultation. Physicians on both sides indicated that they were able to gain knowledge in a useful way by conducting the TCs in 97.9% (*n* = 93 consultations). See also Table 3 for the corresponding data. Additionally, the statement “talking with the telephysician helped in coping with the current situation” was scored from 1 (agree) to 6 (disagree) by the external physicians after each meeting. A grade of 1 was attributed to 85.2% (*n* = 69/81), a grade of 2 to 9.9% (*n* = 8), a grade of 3 to 3.7% (*n* = 3), and a grade of 4 was scored once (1.2%). The two lowest grades, 5 and 6, were both not assigned. A corresponding bar chart is given in Figure 4.

### 3.2. Second Study Section: TCs without Direct Patient Reference

Besides the patient-related meetings, there were also 43 meetings without direct reference to certain patients in the previously presented sample that were solely held between PC physicians. These meetings lasted an average of 22.6 min (SD: 14.7) and were joined by colleagues from other departments in 3 cases (7%). Technical problems occurred in 23.3% (*n* = 10) of such meetings for both physicians and in 9.3% (*n* = 4) for one physician. These TCs were used for educational purposes in 60.5% (*n* = 26), for intercollegiate case discussions in 34.9% (*n* = 15), and for a mixture of both in 4.7% (*n* = 2) of the meetings. In this context, the most frequently mentioned subjects of the intercollegiate case discussion were pain (23.5%; *n* = 4/17) and “miscellaneous” (64.0%; *n* = 11/17), which included diverse topics such as ethical case discussions, treatment goal definition, COVID-19, and cannabis. Participating physicians indicated that they gained knowledge through these conversations in 97.7% (*n* = 42/43 evaluations) of the meetings and rated their discussions with the highest grade of 1 (agree) in 77.5% (*n* = 31/40 external evaluations) of the meetings. However, a grade of 5 was also assigned in four cases (10%), and the lowest grade of 6 was given in two cases (5%).

## 4. Discussion

Several studies have already confirmed that PC and telemedicine are not only compatible but also beneficial in daily clinical practice [17,25,26]. As our findings suggest, TCs may help to conserve the often limited physical resources of patients receiving PC by decreasing or completely bypassing burdensome transfers to university hospitals, even if other specialized departments are required for specific issues. These results are consistent with previously published research showing both high feasibility and acceptance of telemedicine in PC, now also being increasingly applied in more urbanized areas following contact restrictions related to COVID-19 [27,28]. Needless to say, this phenomenon was not only relevant in our study or in the field of PC, but has affected the clinical practice in virtually all medical subspecialities [29,30]. However, if a face-to-face presentation becomes imperative, it is reasonable to assume that the bond and feeling of responsibility for the patient’s well-being will be strengthened by an already established patient-physician relationship and/or physician-physician information exchange online. This hypothesis is backed by studies such as that of Badini et al. from 2022, which demonstrated no differences in ratings of “feeling heard and understood” between patients using telehealth compared with standard care. Moreover, several benefits, including increased efficiency and the ability to involve family members, were also highlighted by PC professionals in this study, matching our findings [31]. Thus, in some cases, our participants reported that both sides were able to improve their planning and preparation for the consultations. Based on our impressions, we also assume that TCs shortened waiting times and thereby improved the comfort for the attending physicians, their patients, as well as their relatives. Owing to the absence of a control group, we are unable to verify this observation conclusively, although similar effects were reported in comparable research [32]. In addition, telehealth has the potential to significantly decrease missed appointments in PC [33]. By using software features to access extended data about patients remotely, the specialized colleagues at the university hospital were able to acquire more detailed insights into the medical records of the targeted patients beyond the doctor’s letters and images that are usually provided. As a result, this could help in reducing loss of information as well as redundancy in the future. Within the context of PC, this is of particular relevance since the timely delivery and completeness of discharge or transfer letters sometimes pose difficulties among patients with limited life expectancies, for which oral transmissions may offer valuable assistance [34,35].

Our results strongly reveal that both outpatient consultations and transfers to the university hospital were partly avoided due to collaborative work between the physicians, which is expected to lower healthcare costs. Reviews corroborated similar impacts [36,37]. Apart from preventing unnecessary presentations, TCs also allow for selective referrals to university hospitals or the mitigation of patients’ preconceptions. For instance, there was a female patient in our cohort who decided to make an appointment at the university hospital based on her familiarity with the attending physicians during the TCs that reduced her general reservations about university hospitals, which were formerly perceived as being impersonal and too crowded. Additional beneficial effects of TCs have also been documented in rural healthcare institutions. For instance, in one study with a small sample size, positive effects were even reported regarding the transfer of critically ill patients who had previously been in contact with PC via telemedicine [38]. With regard to our study results, consideration ought also to be given that in both sections, “miscellaneous” represented by far the most frequent reason for requests for TCs. Although some of these cases are “otherwise unclassified” according to our categories or hide very demanding underlying issues, we also assume that numerous minor problems were covered nonetheless. Potentially, this will result in both personnel and financially valuable economizations within the healthcare system.

The initialization of the collaborations for this study was based on varying degrees of personal acquaintanceship between the external and internal physicians. Thus, all participants were highly interested in an intensified intercollegiate dialogue and bidirectional knowledge transfer. This may also partly contribute to the overall very positive evaluation of the TCs, along with the frequent indication of knowledge gained due to the TCs. Even if the results are judged under this potential bias, this model can nonetheless be regarded as a feasible and convincing approach for an increased exchange in PC [39]. It is worth noting that, in addition to those TCs with specific patient references, there were also several meetings organized by the participating physicians whose purpose was to address entirely unrelated topics, such as continuing education. Based on this, we conclude that, on the one hand, there is an unmet need for PC-related knowledge exchange at smaller hospitals, while, on the other hand, this need could potentially be met through low-threshold services offered by academic hospitals. In the current research, this phenomenon is often referred to as “knowledge mobilization” as a generic term for the process of collecting and sharing research-based knowledge in the health and social care systems [40]. Initial approaches to such knowledge transfer have also been developed in PC [41]. Since positive effects of interprofessional collaboration were demonstrated for patients in PC, one particularly interesting aspect of this study was the opportunity to include other departments of the university hospital, which participated in over 26% of all TCs [42]. Similar results were also demonstrated for other professions not included in this study, such as pharmacists, who may also be more easily accessible within the academic environment [43]. However, there were also a few cases in which the external hospitals’ PC physicians were not interested in any telemedical exchange. These included reservations about meddling by tertiary care professionals into the activities of external hospitals, or the reluctance in using new technology. As a result, one hospital’s technicians rejected their participation in this project and doubted its general potential benefits. Given that both scientific evidence and our findings indicate that the latter statement is outright erroneous, integration barriers necessarily need to be more effectively addressed on an educational level. Yet, there may have also been other non-verbalized reasons for rejection according to prior research, such as the need for staff training, telemedicine as not being a preferred modality, costs, and concerns about low reimbursement [44]. Among other things, this may also be related to the fact that the lack of a standard legal framework both in Germany and worldwide still raises doubts about the protection of patients’ privacy and about liability insurance for the healthcare personnel involved in their treatment [45,46].

What is not apparent from the results of our study are the quite specific reasons for requests and the resulting conversations within TCs in some cases. Therefore, it is not surprising that the most frequent reason for requests was categorized as “miscellaneous”. In this context, legal or ethical considerations may quickly arise for patients receiving PC who need to be counseled by the physicians [47,48]. This is why we present three highly memorable and partly challenging scenarios within TCs in Figure 5 and attempt to derive some of our lessons learned. Moreover, it was striking that, in many cases, several reasons for requests were indicated simultaneously, revealing patients’ symptom complexity despite guideline-based treatment. This matches the high number of unmet needs among patients suffering from advanced diseases, as reported in previous investigations [49]. It also corresponds to observations suggesting that, during this phase of disease, patients often experience multiple symptoms. Thus, Bausewein et al. observed that patients diagnosed with advanced cancer or chronic obstructive pulmonary disease presented with an average of 14 symptoms [50].

While conducting this study, it became apparent that a fundamental prerequisite for successful TCs is the availability of adequate technical infrastructure. Despite appropriate training and the provision of suitable hardware and software, technical issues were observed in one-third of all meetings for at least one physician. At present, this deficiency seems to be quite prevalent in telemedicine in general and was also reported in the gynecology-obstetrics arm of our parent study [22,45]. As an example, one of our collaborating hospitals lacked wireless internet coverage throughout the building. Moreover, even if a connection was established, this was still insufficient for TCs to use video functions. Beyond that, education, and readily available technical support by trained personnel, and the participants’ technical expertise are of great importance, allowing inexperienced staff to become familiar with new technologies and to be instructed remotely [51]. Heading into the future, we endorse that large-scale implementation of telehealth thus necessitates further research, a greater emphasis on training, the establishment of appropriate telehealth capacity, assured financial viability, clear legislative frameworks, and deepened collaboration with the technology industry [52].

### Limitations

In total, the present study comprises only a comparatively small number of included participants and conducted TCs. This was mainly caused by the outbreak of COVID-19, which started shortly after the study launch and persisted up until its completion. Therefore, this aspect can be considered both a drawback and an advantage. Thus, while the number of patients in hospitals was markedly lower compared to pre-pandemic levels, thereby impeding recruitment, patients were also more likely to be open toward remote medical concepts. In this context, it is also important to note that all of the contributing physicians were already overburdened due to commonly known circumstances and additional responsibilities following the pandemic, leaving only limited resources for research activities. Moreover, it is important to consider that only TCs between physicians were investigated in this study. Although some of the meetings involved other professions, such as social work, further benefits might have been gained by including other professions (e.g., nursing, spiritual care, or psychiatry) in a more standardized approach. Potentially, our selected study design may also have left some patients in need of specialized PC unrecognized since only the inpatient primary care physicians selectively identified patients. A more comprehensive screening would have been preferable here but was not feasible given the increased workload due to COVID-19 and the overall shortage of staff in hospitals. Similarly, qualitative interviews as part of a mixed-methods framework would have supported and illustrated our findings in more detail. Again, this had to be abandoned owing to limited resources. Regardless of COVID-19, one obvious shortcoming of this study is the absence of a control group, but this was incorporated in a different yet unpublished arm of its parent study. However, due to the focus on knowledge exchange and intercollegiate meetings, such an approach would most likely yield only a few benefits with enormous additional effort. Also, as has been described above, technical problems occurred in several TCs, potentially influencing the evaluations and outcomes of this study. As a result, in a few cases, complete data sets could not be retrieved for all patients or TCs. Additionally, a very probable bias can be found because most participating physicians knew each other beforehand. For this reason, it is conceivable that these participants tended not to rate poorly. Notwithstanding the predominantly good ratings, there were also a few very poor ratings, indicating that such an effect may not have affected all physicians. Conversely, a major advantage of involving hospitals and physicians already familiar with each other was the increased willingness to participate, which led to the inclusion of five primary care hospitals besides the tertiary care hospital. Weighing this potential bias against the possibility of a larger multicenter study, we decided on the latter option for this study. Finally, it needs to be borne in mind that this study was solely conducted in a highly developed region with easy access to technical resources and the internet (at least on paper). Hence, the extent to which our findings are transferable to less developed or culturally divergent regions remains questionable.

## 5. Conclusions

Telemedicine enables the transfer of university expertise to regional hospitals and improves collaborative care while maintaining high satisfaction among physicians. Based on our findings, we were able to demonstrate that TCs may also avoid noteworthy numbers of transfers and outpatient presentations to tertiary care hospitals, leading to potentially substantial cost savings within the healthcare system. In doing so, it is feasible to counsel a wide variety of reasons for the request in a purposeful manner, ranging from minor concerns to challenging and more complex situations that may result in subsequent selective transfers. Nonetheless, we strongly recommend the use of appropriate digital infrastructure as essential for connecting external patients to tertiary care hospitals at their bedside in a purposeful manner. Thus, although appropriately equipped, technical problems were encountered in one-third of all TCs in this study. Yet, there are also still reservations regarding both the use of technology and university co-care in primary care hospitals. Consequently, educational programs covering the benefits, potential risks, and limitations of this collaborative care approach need to be offered throughout the healthcare landscape, backed up by a sound financial basis for telemedicine in PC.

## Figures and Tables

**Figure 1 cancers-15-02512-f001:**
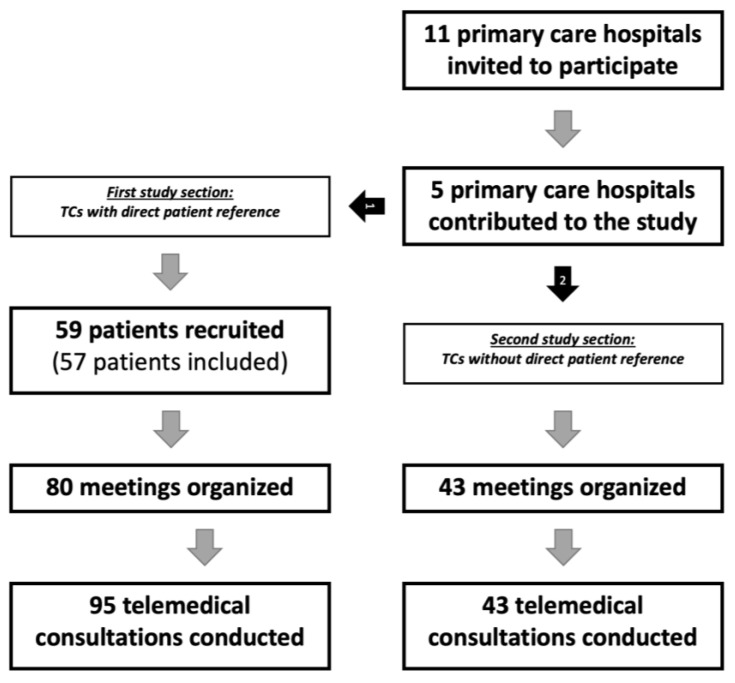
Flowchart of the recruitment process (period: January 2020 to October 2021). Abbreviations: TCs = Telemedical consultations.

**Figure 2 cancers-15-02512-f002:**
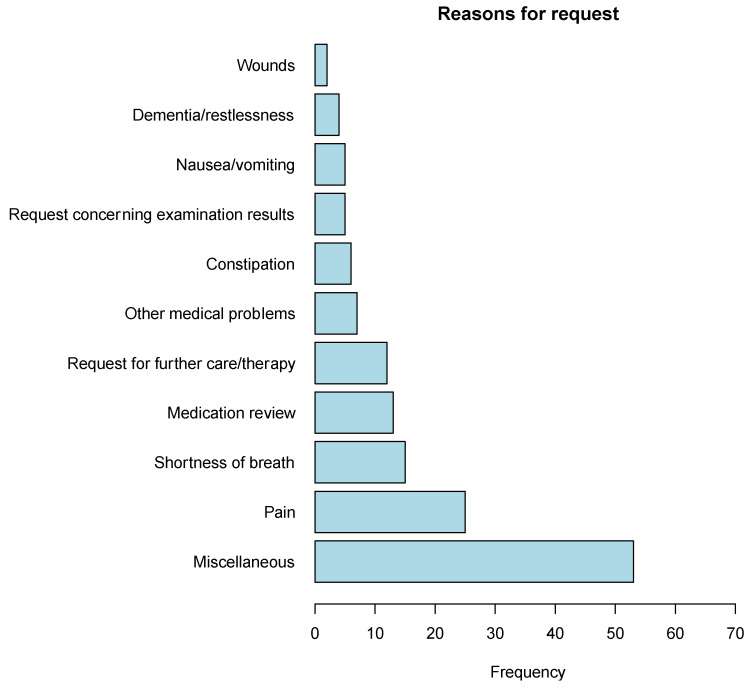
Frequencies of reasons for requesting telemedical consultations (absolute numbers related to data of *n* = 71 specifications out of *n* = 95 consultations; several reasons could be given).

**Figure 3 cancers-15-02512-f003:**
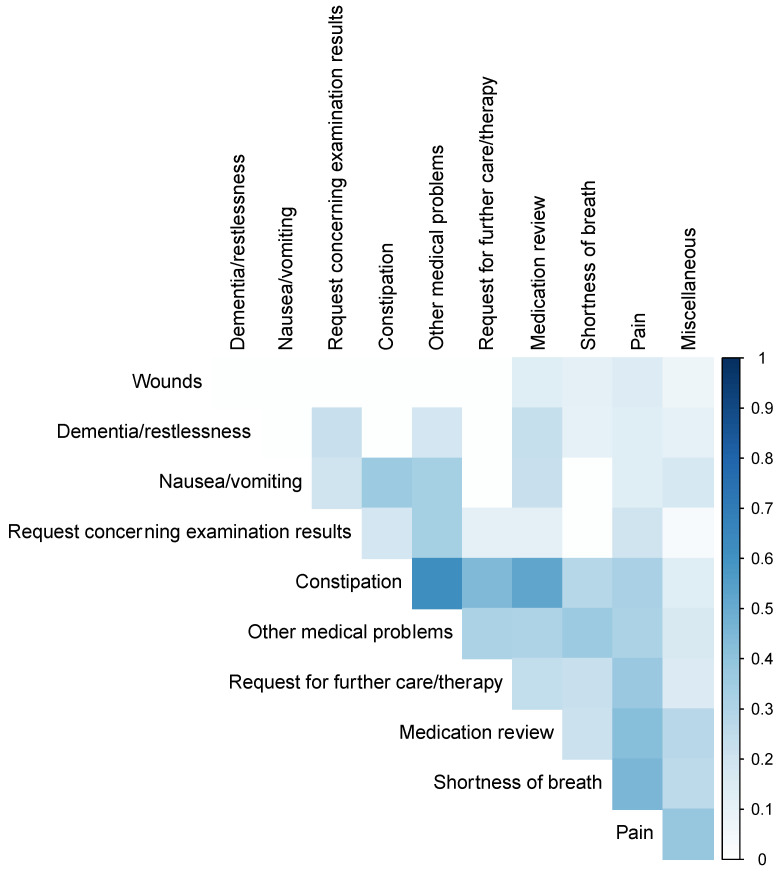
Visualization of association between the reasons for request. For this purpose, it was analyzed how frequently two reasons were mentioned simultaneously compared to their general frequency (related to data of *n* = 71 specifications out of *n* = 95 consultations; method: coincidence index by Dice).

**Figure 4 cancers-15-02512-f004:**
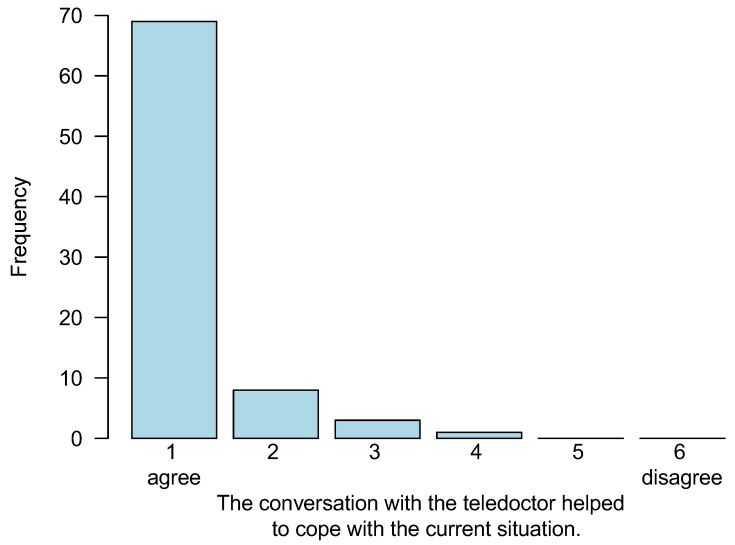
Frequencies concerning physicians’ ratings of help from the telemedical consultations based on an ordinal scale ranging from 1 (agree) to 6 (disagree) (absolute numbers, related to *n* = 81 evaluations by the external physician out of *n* = 95 meetings).

**Figure 5 cancers-15-02512-f005:**
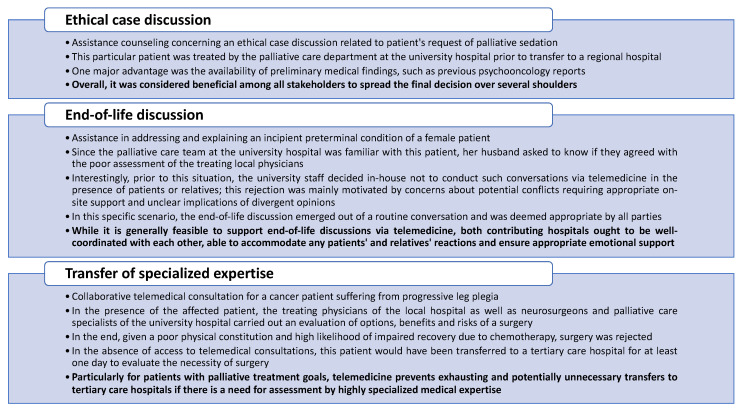
Three examples of particularly outstanding scenarios encountered during telemedical consultations, the way we managed them, and some of our lessons learned.

**Table 1 cancers-15-02512-t001:** Patient, clinical, and care characteristics (*n* = 57). Abbreviations: SD = Standard deviation; km = Kilometer; GSSC = German social insurance classification (smaller numbers = lower need for help); COPD = Chronic obstructive pulmonary disease.

**Age (Years)**	
Min/max Mean (SD)	27.1/94.3 69.7 (13.6)
**Sex**	
Female, *n* (%) Male, *n* (%)	34 (59.6%) 23 (40.4%)
**Distance to clinic (km)**	
Min/max Mean (SD)	0/102.0 20.6 (21.6)
**Marital status**	
Married/partnered, *n* (%) Single, *n* (%) Divorced, *n* (%) Widowed, *n* (%) Not known, *n* (%)	34 (59.6%) 6 (10.5%) 3 (5.3%) 12 (21.1%) 2 (3.6%)
**Main diagnosis**	
Cancer (total), *n* (%) Cancer (metastasized), *n* (%) COPD, *n* (%) Other, *n* (%)	46 (80.7%) 39 (68.4%) 3 (5.3%) 8 (14.0%)
**Phase of illness**	
Stable, *n* (%) Unstable, *n* (%) Deteriorating, *n* (%) Dying, *n* (%) Not known, *n* (%)	13 (22.8%) 24 (42.1%) 16 (28.1%) 1 (1.8%) 3 (5.3%)
**Active treatment**	
Yes, *n* (%) No, *n* (%) Not known, *n* (%)	21 (36.8%) 27 (47.4%) 9 (15.8%)
**Type of active treatment**	
Radiation therapy, *n* (%) Chemo- or immunotherapy, *n* (%) Surgery, *n* (%)	13 (22.8%) 25 (43.9%) 21 (36.8%)
**Defined treatment limits**	
Yes, *n* (%) No, *n* (%) Not known, *n* (%)	27 (47.4%) 23 (40.4%) 7 (12.3%)
**Days of treatment**	
Min/max Mean (SD)	1.0/56.0 12.5 (13.7)
**Care dependency (GSSC)**	
No need for care, *n* (%) Level of care 1, *n* (%) Level of care 2, *n* (%) Level of care 3, *n* (%) Level of care 4, *n* (%) Level of care 5, *n* (%) Not known, *n* (%)	11 (19.3%) 0 (0.0%) 8 (14.0%) 13 (22.8%) 1 (1.8%) 0 (0.0%) 24 (42.2%)

**Table 2 cancers-15-02512-t002:** General session characteristics of the telemedical consultations (*n* = 95 different consultations within *n* = 80 meetings); * decided reasons for requests of these three categories can be found in Figure 2. Abbreviations: SD = Standard deviation.

**Duration (*n* = 80)**	
Min/max Mean (SD)	0/73.0 23.1 (14.7)
**Involvement of further medical departments (*n* = 80)**	
Yes, *n* (%) No, *n* (%) Not known, *n* (%)	21 (26.2%) 57 (71.2%) 2 (2.5%)
**Involved medical departments**	Dermatology, Oncology, Gynecology, Infectiology, Neurology, Neurosurgery, Gastroenterology, Pneumology, Radiotherapy, Urology, Social Service
**Purpose of the initial consultation request (related to *n* = 57 patients)**	
Palliative medicine consultation *, *n* (%) Patient transfer, *n* (%) Intercollegiate case conference *, *n* (%) Symptoms or administration *, *n* (%) Education, *n* (%) Not known, *n* (%)	22 (38.6%) 17 (29.8%) 12 (21.2%) 4 (7.0%) 1 (1.8%) 1 (1.8%)
**Occurrence of technical issues (*n* = 80)**	
Yes (both physicians), *n* (%) Yes (one physician), *n* (%) No, *n* (%)	16 (20.0%) 13 (16.2%) 51 (63.7%)
**Type of technical issues**	Login, video quality, sound quality, session establishment, data exchange, session termination, internet connection difficulties, and consultation requests not received
**Consultations in the presence of the patient (*n* = 95)**	
Yes, *n* (%) No, *n* (%) Not known, *n* (%)	34 (35.8%) 59 (62.1%) 2 (2.2%)
**Consultation results (*n* = 71)**	
Adjustment of therapy or medication No further measures Other unclassified usable results	25 (35.2%) 13 (18.3%) 33 (46.5%)

**Table 3 cancers-15-02512-t003:** Avoidance of outpatient medical appointments, transfers, and gain of knowledge due to the telemedical consultations (*n* = 95 different consultations within *n* = 80 sessions); * “contradictory” means that the statements of the two physicians concerned are not in agreement.

**Avoided Outpatient Presentations (*n* = 95)**	
Yes, *n* (%)	20 (21.1%)
No, *n* (%)	3 (3.2%)
Contradictory, *n* (%) *	15 (15.8%)
Not relevant, *n* (%)	57 (60.0%)
**Avoided Transfers of Patients (*n* = 95)**	
Yes, *n* (%)	12 (12.6%)
No, *n* (%)	9 (9.5%)
Contradictory, *n* (%) *	18 (19.0%)
Not relevant, *n* (%)	56 (58.9%)
**Gain of Knowledge (*n* = 95)**	
Yes, *n* (%)	93 (97.9%)
No, *n* (%)	0 (0.0%)
Contradictory, *n* (%) *	2 (2.1%)

## Data Availability

The data presented in this study are available on request from the corresponding author. The data are not publicly available due to reasons of data protection.

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
