# Peer review of "Telemedical Consultations in Palliative Care: Benefits through Knowledge Exchange and Intercollegiate Collaboration—Findings from the German oVID Project"

_cancers, 2023, doi:10.3390/cancers15092512_

Round 1

Reviewer 1 Report

Authors reported a prospective multi-center feasibility trial about the role of telemedical consultations in palliative care. This topic is interesting, look at these points carefully to improve it:

- Lines 95-98: "As can be seen from the current scientific status, beside immediate therapeutic and care benefits for individual... areas were addressed and evaluated in this study" What is the purpose of this paper? State it clearly here.

- In the results, it seems that "miscellaneous" represents a very important part (64.0%) of the requests for telemedical consultations. The use of telemedicine for "minor" reasons can certainly be an advantage for the entire hospital system, also in terms of health care expenditure. What do the authors think?

- Lines 260-268: "related to COVID-19 [24,25]. " It is important to report that Covid-19 probably changed clinical practice in all medical subspecialities. Consider refs. -- DOI: 10.1080/02688697.2020.1773399  --  DOI: 10.3389/fcimb.2021.749911 -- DOI: 10.1007/s00345-020-03333-6

- Lines 238-240: Figure 4. An important point is "How do consumers prefer their care delivered: In-person, telephone or videoconference?" -- DOI: 10.1177/1357633X231160333  --  DOI: 10.1186/s13643-017-0529-0

- Lines 333-340: "Yet, there may have also been other non-verbalized reasons for rejection according to prior...  reimbursement [39]." Is it important to report that "the lack of a standard legal framework causes some doubts about patient privacy, liability coverage for treating healthcare workers" as also financial reimbursements. Look at refs. DOI: 10.1177/21925682221090891  -- DOI: 10.7326/0003-4819-129-2-1998

- Lines 384-388: "Additionally, a very probable bias can be found because most participating physicians had known each other beforehand. For this reason, it is conceivable that these participants tended not to rate poorly" What do author propose?

- Lines 394-402. Improve conclusion section. What does it add new to the literature? Report here some results and conclusion of this paper.

- Figure 1 can probably be improved by adding more data

Minor editing of English language required

Author Response

Authors reported a prospective multi-center feasibility trial about the role of telemedical consultations in palliative care. This topic is interesting, look at these points carefully to improve it:

- Lines 95-98: "As can be seen from the current scientific status, beside immediate therapeutic and care benefits for individual... areas were addressed and evaluated in this study" What is the purpose of this paper? State it clearly here.

Ad 1) We have adapted and specified this accordingly (p. 3, ll. 149-156).

- In the results, it seems that "miscellaneous" represents a very important part (64.0%) of the requests for telemedical consultations. The use of telemedicine for "minor" reasons can certainly be an advantage for the entire hospital system, also in terms of health care expenditure. What do the authors think?

Ad 2) We generally agree with this statement. However, in the context of our study and palliative care as a whole, it should be borne in mind that a clear categorization is difficult and that " miscellaneous" may encompass not only minor topics but also highly relevant and complex topics, particularly in the context of consultations relating to direct patients. Nevertheless, we consider this point to be very important and have added it to our discussion (p. 12, ll. 389-395).

- Lines 260-268: "related to COVID-19 [24,25]. " It is important to report that Covid-19 probably changed clinical practice in all medical subspecialities. Consider refs. -- DOI: 10.1080/02688697.2020.1773399  --  DOI: 10.3389/fcimb.2021.749911 -- DOI: 10.1007/s00345-020-03333-6

Ad 3) We have briefly included this notion into the manuscript (p. 11, ll. 351-353).

- Lines 238-240: Figure 4. An important point is "How do consumers prefer their care delivered: In-person, telephone or videoconference?" -- DOI: 10.1177/1357633X231160333  --  DOI: 10.1186/s13643-017-0529-0

Ad 4) Although we do agree with this comment, we did not address it anew in our study since there is already a considerable amount of research available, as your literature recommendations also demonstrate. Besides, we have already briefly mentioned this point in the introduction (p. 2, ll. 104-110).

- Lines 333-340: "Yet, there may have also been other non-verbalized reasons for rejection according to prior...  reimbursement [39]." Is it important to report that "the lack of a standard legal framework causes some doubts about patient privacy, liability coverage for treating healthcare workers" as also financial reimbursements. Look at refs. DOI: 10.1177/21925682221090891  -- DOI: 10.7326/0003-4819-129-2-1998

Ad 5) We have embedded this important information in the manuscript (p. 12, ll. 426-429). Thank you!

- Lines 384-388: "Additionally, a very probable bias can be found because most participating physicians had known each other beforehand. For this reason, it is conceivable that these participants tended not to rate poorly" What do author propose?

Ad 6) From our point of view, the most transparent and comprehensible approach here is the one we have already chosen: to name this bias quite openly and honestly. Given that some of the participating physicians are also authors of this article, any other form of assessment of this aspect appears scientifically inappropriate to us. We are, of course, open to any better proposals for this problem. Otherwise, we believe that, although we are not able to prevent this bias, at least we are able (and morally and scientifically committed) to clearly indicate it. In this context, it is also important to bear in mind that, by accepting this bias, the number of participating hospitals was substantially increased (p. 14, ll. 495-499).

- Lines 394-402. Improve conclusion section. What does it add new to the literature? Report here some results and conclusion of this paper.

Ad 6) We have revised this section accordingly (p. 14, ll. 504-518).

- Figure 1 can probably be improved by adding more data

Ad 7) We have added some additional data (p. 5).

We thank Reviewer 1 for his/her comments. His/Her points have substantially improved the revised version of our manuscript.

Reviewer 2 Report

This is a very interesting and important topic, particularly since the COVID-19 pandemic telemedicine has increased a lot.

However, I am a bit  confused. Not before the methods section, and otherwise than the title of the manuscript suggests, it becomes clear that the intervention does not (merely) concern consultations between patients and health care professionals, but mainly consultations between palliative care experts and other health care professionals (from other hospitals? not GPs?). Only in a part of the consultations a patient is present, and only in a part of the consultations it concerns one or more specific patients. 

In fact, this means, if I interpret correctly, that this study is a combination of telemedicine consultations with or about a patient, telemedicine consultations with colleagues without a patient, and interprofessional education. 

This really needs to be made clear in the introduction ánd in the title. I would prefer that you delete the 'consultations' that concerned education (not referring to a specific patient), as that is a totally different aim than consultations regarding patients.

Next: I miss studies of van Gurp J et al and of Hoek P about teleconsultations in palliative care

Minor remarks:

Introduction: 

page 2, line 69: increased survival was only found in some studies. As many studies (reviews) show no prolonged length of survival. I would suggest to mention that PC does not DECREASE survival.

Page 2, line 87: what do you mean with 'direct communication'? Triggered by this: as PC concerns four dimensions (physical, social, psychological and spiritual): I am really surprised that only physical symptoms are mentioned. Did the recommendations also concern the other dimensions?

table 2: check the lay out (the different columns do not match everywhere)

Page 8, line 223: how did you measure that face to face presentations were prevented? And wouldn't it be logical that such expertise consultations would also increase referrals?

Author Response

This is a very interesting and important topic, particularly since the COVID-19 pandemic telemedicine has increased a lot.

However, I am a bit confused. Not before the methods section, and otherwise than the title of the manuscript suggests, it becomes clear that the intervention does not (merely) concern consultations between patients and health care professionals, but mainly consultations between palliative care experts and other health care professionals (from other hospitals? not GPs?). Only in a part of the consultations a patient is present, and only in a part of the consultations it concerns one or more specific patients. 

In fact, this means, if I interpret correctly, that this study is a combination of telemedicine consultations with or about a patient, telemedicine consultations with colleagues without a patient, and interprofessional education. 

This really needs to be made clear in the introduction ánd in the title. I would prefer that you delete the 'consultations' that concerned education (not referring to a specific patient), as that is a totally different aim than consultations regarding patients. 

Ad 1) Although we appreciate your concerns, we assess this topic in a quite different way. As the proposed title "Telemedical consultations in Palliative Care: Benefits through knowledge exchange and intercollegiate collaboration-Findings from the German oVID Project" very clearly indicates, the focus of the study is on an interprofessional and not on a physician-patient level. Similarly, we emphasize the goal of "transfer university expertise" in the abstract rather than the aim of direct patient care. Additionally, the physician's perspective was primarily investigated, which is why we explicitly state that patients were not necessarily present throughout. Furthermore, of all telemedical consultations conducted within this project, over two-thirds were directly related to individual patients, so we consider the selected title to be most appropriate for the present study. Your confusion may be due to the fact that medical consultations do not have congruent characteristics in different countries. Thus, in Germany, the patient does not have to be present, nor does the consultation have to take place on-site. Theoretically, it may even take place via telephone. According to the English Cambridge dictionary (https://dictionary.cambridge.org), our impression is that you are applying the term "consultation" much more narrowly than it is (or at least can be) actually used. In our understanding, it is also regularly used in the context of legal or political issues in the sense of "consulting", "advice" or "counseling". Therefore, we would like to refrain from amending the title, but instead have revised the "Simple Summary" and the "Abstract" to make them more precise with regard to this study, its methods and aims (p. 1, ll. 22-48). In this context, we are also supported by your literature recommendation in the subsequent comment, where Gurp et al. also refer to the term "teleconsultations" and mention that "In 17/18 cases, interprofessional contact was restricted to backstage work after teleconsultation". In our opinion, no linguistic distinction between consultations with or without patients can and should be made in this context, since even in meetings that are rather oriented towards education, there are always specific current or past patient cases that are discussed, as supervision and exchange are incredibly effective and important resources, particularly in fields such as palliative care. We also do not fully understand your point of criticism regarding the target group ("(from other hospitals? not GPs?") since it is mentioned explicitly both in the Simple Summary and in the Abstract sections. This study investigated collaboration exclusively between inpatient hospital-based physicians who practiced either in tertiary care or primary care settings. For your utmost satisfaction, however, we also emphasized this aspect to a greater extent in the “Simple summary” and “Abstract” (p. 1, ll. 22-48).

Next: I miss studies of van Gurp J et al and of Hoek P about teleconsultations in palliative care

Ad 2) Many thanks for your hint! We have now integrated one study from each of the two authors in the introduction (p. 2, ll. 131-135).

Minor remarks:

Introduction: 

page 2, line 69: increased survival was only found in some studies. As many studies (reviews) show no prolonged length of survival. I would suggest to mention that PC does not DECREASE survival.

Ad 3) We have revised the statement accordingly (p. 2, l. 117).

Page 2, line 87: what do you mean with 'direct communication'? Triggered by this: as PC concerns four dimensions (physical, social, psychological and spiritual): I am really surprised that only physical symptoms are mentioned. Did the recommendations also concern the other dimensions?

Ad 4) Since these are the findings from Dillen et al.'s study of 2020 and we are solely quoting their main results here, a comprehensive explanation of this can be found here: DOI: 10.1017/S147895152000125X. Second point: We are not entirely sure which recommendations you are exactly referring to. As we have stated several times in our manuscript, a wide variety of reasons for consultations occurred among patients and physicians during this study, of which, as an example, three particularly specific situations are described in greater detail in Figure 5 (p. 13). Therefore, not only physical symptoms were taken into account in our study, which—you are absolutely right—would not be appropriate for palliative approaches. However, we have now mentioned psychological symptoms explicitly in the Introduction so that their significance in palliative care is emphasized more clearly in our paper from the very beginning (p. 2, l. 122).

table 2: check the lay out (the different columns do not match everywhere)

Ad 5) Thank you for this tip! However, this table (p. 8-9) is displayed completely accurately in Microsoft Office used on multiple devices so, unfortunately, we cannot reproduce this comment from our technical point of view. However, we assume that a potential shift would catch our attention during the last inspection of the PDF prior to its publication so that we will then pay closer attention to this comment once again. This error may also depend on differences in the version given to the reviewers or the text editing program being used so in case of doubt we will contact the editors once again.

Page 8, line 223: how did you measure that face to face presentations were prevented? And wouldn't it be logical that such expertise consultations would also increase referrals?

Ad 6) The telemedical consultations with direct patient reference were conducted as a result of specific problems or questions arising among the physicians at the external hospitals. Without a sufficient solution through the telemedical consultations, transfers or outpatient presentations would have been recommended by the external colleagues so that the two physicians concerned could evaluate the prevention of these accordingly. However, we supplemented in the methods section that this assessment was made jointly by the primary care and tertiary care physicians involved after each meeting (p. 4, ll. 218-220). Second point: We have already briefly addressed the option of selective referrals in the discussion (p. 12, ll. 381-382). To the best of our knowledge, there are no studies on the effect of the increase in referrals by now. Yet, we do not expect your hypothesis to be true according to our experience at a hospital running multiple telemedical studies since the number of avoided referrals far exceeds the number of additional selective referrals in our opinion.

We also thank Reviewer 2 for his/her valuable comments. His/Her points have substantially improved the revised version of our manuscript.

Reviewer 3 Report

Thank you for an interesting study on telemedicine al consultations.

I have some comments and suggestions for improvement:

1) To me it is not clear why you have only used physicians as informants. PC is usually a team approach including at least both physicians and nurses / a multidisciplinary team. Although the physicians views are important you should explain a bit more why you chose solely physicians. This should also be included in the limitations section.

2) A bit more informations about the method of including patients or not would be useful for the reader. Were all patients asked to participate or was it just up to the physician to make this choice. This should also be part of the discussion because it is a major issue in patient-centred care. 

3) In the method section I miss more detailed information about the questionnaire. Could it be added as supplementary material?

4) The findings could have been strengthened by using qualitative interviews with the physicians. Has this been considered? Please discuss. 

5) Is your primary outcome appropriate for video meetings that are not patient case related?

6) Fig. 3 might be complicated to read and understand for some readers. Are there other ways to present these results to make them easier understandable?

Best wishes for your revision of the paper!

Author Response

Thank you for an interesting study on telemedicine al consultations.

I have some comments and suggestions for improvement:

1) To me it is not clear why you have only used physicians as informants. PC is usually a team approach including at least both physicians and nurses / a multidisciplinary team. Although the physicians views are important you should explain a bit more why you chose solely physicians. This should also be included in the limitations section.

Ad 1) We have now explained this in greater detail in the methods (p. 3, ll. 178-184) and limitations (p. 14, ll. 474-482) sections. As we are talking about a transfer of knowledge from a tertiary care hospital to a primary care hospital, anything other than the inclusion of the most highly qualified staff (which is usually the physician in a medical setting for many and, in particular, very specific issues) would not be useful. To ensure that other qualifications are not left out, however, we have intentionally provided the opportunity to integrate other specialized disciplines at any time (p. 3, ll. 183-184). 

2) A bit more informations about the method of including patients or not would be useful for the reader. Were all patients asked to participate or was it just up to the physician to make this choice. This should also be part of the discussion because it is a major issue in patient-centred care. 

Ad 2) We have expressed this information a little more precisely (p. 4, ll. 213-215).

3) In the method section I miss more detailed information about the questionnaire. Could it be added as supplementary material?

Ad 3) Since the questionnaire (i) contains numerous conditional questions, (ii) has two versions (university hospital and primary care provider), and (iii) is drafted in German, we do not consider a general embedding in the "supplementary materials" section to be expedient but would rather consider a distribution of these documents upon request to be more appropriate, as mentioned in the "Data availability statement" (p. 15, ll. 542-543). We are presenting the outcomes of the main issues in the results and think it would be preferable for any specific queries to be directed to the corresponding author so that issues concerning single items, or the questionnaire as a whole, can be discussed in more detail than can be done in this article.

4) The findings could have been strengthened by using qualitative interviews with the physicians. Has this been considered? Please discuss. 

Ad 4) We have now briefly commented on this in the Limitations section. We strongly agree with your statement, but due to project-related third-party funding and limited human resources, we were not able to integrate a mixed-methods approach including a qualitative component into this project although it would undoubtedly be beneficial (p. 14, ll. 482-484).

5) Is your primary outcome appropriate for video meetings that are not patient case related?

Ad 5) Thank you for pointing this out. The primary outcome only refers to the patient-referred cases. We have adjusted this accordingly in the Methods section (p. 4, ll. 216-227).

6) Fig. 3 might be complicated to read and understand for some readers. Are there other ways to present these results to make them easier understandable?

Ad 6) We agree with you on this point, but for this very reason Figure 2 (p. 7) and Figure 3 (p. 8) are based on the same data so that Figure 3 hopefully also can easily be understood by even inexperienced readers with the help of Figure 2 (albeit without their respective association to each other) showing frequencies as a simple bar chart. Moreover, from our point of view, Figure 3 is primarily interesting for other researchers for whom these additional data might provide important further input.

We also thank Reviewer 3 for his/her valuable comments. With the implementation of the comments of the three reviewers, the manuscript has gained much more quality and accuracy.

Round 2

Reviewer 1 Report

ok

ok